

# The Small Whiskbroom Imager for atmospheric compositioN monitorinG (SWING) and its operations from an Unmanned Aerial Vehicle (UAV) during the AROMAT campaign

Alexis Merlaud[1], Frederik Tack[1], Daniel Constantin[2], Lucian Georgescu[2], Jeroen Maes[1], Caroline Fayt[1], Florin Mingireanu[3], Dirk Schuettemeyer[4], Andreas Carlos Meier[5], Anja Schönardt[5], Thomas Ruhtz[6], Livio Bellegante[7], Doina Nicolae[7], Mirjam Den Hoed[8], Marc Allaart[8], and Michel Van Roozendael[1]

[1]Royal Belgian Institute for Space Aeronomie (BIRA-IASB), Avenue Circulaire 3, 1180 Brussels, Belgium
[2]"Dunarea de Jos" University of Galati, Str. Domneasca 111, Galati 800008, Romania
[3]Romanian Space Agency (ROSA), Mendeleev Street, nr. 21-25, Bucharest 10362, Romania
[4]European Space Agency (ESA-ESTEC), Keplerlaan 1, 2201 AZ Noordwijk, The Netherlands
[5]Institute of Environmental Physics, University of Bremen, Otto-Hahn-Allee 1, 28359 Bremen, Germany
[6]Institute for Space Sciences, Free University of Berlin, Carl-Heinrich-Becker-Weg 6-10, 12165 Berlin, Germany
[7]National Institute of R&D for Optoelectronics (INOE), Street Atomistilor 409, Magurele 77125, Romania
[8]Royal Netherlands Meteorological Institute (KNMI), Utrechtseweg 297, 3731 GA De Bilt, the Netherlands

*Correspondence to:* A. Merlaud (alexism@oma.be)

**Abstract.**

The Small Whiskbroom Imager for atmospheric compositioN monitorinG (SWING) is a compact remote sensing instrument dedicated to mapping trace gases from an Unmanned Aerial Vehicle (UAV). SWING is based on a compact visible spectrometer and a scanning mirror to collect scattered sunlight. Its weight, size, and power consumption are respectively 920 g, 27x12x8 cm$^3$, and 6 W. SWING was developed in parallel with a 2.5 m flying wing UAV. This unmanned aircraft is electrically powered, has a typical airspeed of 100 km h$^{-1}$, and can operate at a maximum altitude of 3 km.

We present SWING-UAV experiments performed in Romania on 11 September 2014 during the Airborne ROmanian Measurements of Aerosols and Trace gases (AROMAT) campaign. The UAV was operated up to 700 m above ground, in the vicinity of the large power plant of Turceni (44.67°N, 23.41°E, 116 m a.s.l.). These SWING-UAV flights were coincident with another airborne experiment using the Airborne imaging Differential Optical Absorption Spectroscopy (DOAS) instrument for Measurements of Atmospheric Pollution (AirMAP), and with ground-based DOAS, lidar, and balloone-borne in-situ observations.

The spectra recorded during the SWING-UAV flights are analyzed with the DOAS technique. This analysis reveals $NO_2$ differential slant column densities (DSCDs) up to 13 ± 0.6 x 10$^{16}$ molec.cm$^{-2}$. These $NO_2$ DSCDs are converted to vertical column densities (VCDs) by estimating air mass factors. The resulting $NO_2$ VCDs are up to 4.7 ± 0.4 x 10$^{16}$ molec.cm$^{-2}$. The water vapour DSCD measurements, up to 8 ± 0.15 x 10$^{22}$ molec.cm$^{-2}$, are used to estimate a volume mixing ratio of water vapour in the boundary layer of 0.013 ± 0.002 mol.mol$^{-1}$. These geophysical quantities are validated with the coincident measurements.



# 1   Introduction

Unmanned Aerial Vehicles (UAVs) are increasingly used in civilian applications, and a large variety of UAV platforms is now available as remote sensing platforms for scientific research (Watts et al., 2012). Regarding atmospheric studies, the first published works using UAVs were mainly performed from two kinds of platforms, as can be seen from the review of Houston

et al. (2012, Table. 1). The first category consists of large aircraft of military origin, with wingspan longer than 10 m, such as the GNAT-750 and Altus platforms used in the pioneer work of Stephens et al. (2000) to study the radiative transfer through clouds, or by Mach et al. (2005) to study thunderstorms. The second category is made of much smaller UAVs, with a wingspan under 3 m. Several authors report meteorological observations in the lower troposphere with small dedicated UAV systems such as the Aerosonde (Holland et al., 2001; Curry et al., 2004; Lin, 2006) the Meteorological Mini UAV M$^2$AV (Spiess et al.,

2007; Martin et al., 2011; van den Kroonenberg et al., 2008) or more recently the Small Unmanned Observer (SUMO) (Reuder et al., 2009; Mayer et al., 2012; Cassano, 2014). Small UAVs have also been used to study the aerosol vertical distribution in the Arctic (Bates et al., 2013). Illingworth et al. (2014) have operated an ozone sonde on a small UAV and Watai et al. (2006) have measured $CO_2$ in the lower troposphere. In this paper, we present a remote sensing instrument dedicated to mapping air pollutants from a small UAV. The capabilities of this new observation system are illustrated by measurements around a power

plant in Turceni (Romania), in August 2014.

Several remote sensing instruments have been described to quantify trace gases from traditional aircraft. Those operating in the UV-visible range often use the Differential Optical Absorption Spectroscopy (DOAS) method (Platt and Stutz, 2008). Early airborne DOAS studies were based on zenith-only measurements from aircraft to quantify the stratospheric $NO_2$ and $O_3$ at high latitudes (Wahner et al., 1989; Pfeilsticker and Platt, 1994). The technique was then modified adding nadir and off-axis

angles (Melamed et al., 2003; Wang et al., 2005; Bruns et al., 2006) to study the troposphere. Measuring simultaneously in several lines of sight or in limb geometry during the ascent and descents of the planes, enable to retrieve some information on the vertical distribution of the investigated absorber. This was achieved with various instruments in several campaigns for $NO_2$ but also $O_4$, IO, BrO, formaldehyde, and glyoxal (Merlaud et al., 2011; Prados-Roman et al., 2011; Baidar et al., 2013; Dix et al., 2016). Regarding the atmospheric horizontal distribution, a growing interest appeared in the last decade for nadir-

looking airborne DOAS instruments able to map trace gases below the aircraft. Maps of tropospheric $NO_2$ were acquired around power plants in South Africa (Heue et al., 2008) and in Germany (Schönhardt et al., 2015), or above urban areas like Houston (Kowalewski and Janz, 2009; Nowlan et al., 2016), Zurich (Popp et al., 2012), Indianapolis (General et al., 2014), Brussels and Antwerp (Tack et al., 2017), Leicester (Lawrence et al., 2015), and Bucharest (Meier et al., 2017). General et al. (2014) also quantified $NO_2$ in the vicinity of the Prudhoe Bay oil field in Alaska. Constantin et al. (2017) used a compact

nadir-only set-up to sample the exhaust plume of a steel factory from an ultralight trike in Galati, Romania. These airborne measurements typically yield a spatial resolution of around 100 meters, i.e. two orders of magnitude better than the resolution currently achieved from space (Levelt et al., 2006), and are particularly interesting to study atmospheric chemistry at the local scale and investigate point-source emissions.



The aforementioned airborne DOAS observations were all performed from manned aircraft. Few DOAS observations from UAVs are reported. The ACAM instrument (Kowalewski and Janz, 2009) has been operated from the NASA Global Hawk during the 2010 GloPac campaign. Stutz et al. (2016) report limb-scanning measurements with a new mini-DOAS instrument, also operated from the Global Hawk. Flying above 15 km altitude for a day or more with capacity for heavy payload, this plat-

form has high scientific potential for atmospheric research in general, including DOAS observations. However, it is expensive to operate and logistically demanding. In comparison, smaller UAVs, flying at lower altitudes, present advantages for DOAS studies, especially within the boundary layer. Flying above urban areas is still legally difficult with small UAVs, but, as will be shown below, it is possible to use small UAV to study isolated point sources like a power plant in a rural area. Moreover, the technique could be used for ship emission monitoring, as is already done from manned aircraft (Berg et al., 2012).

In the next section, we describe the SWING payload and the UAV platform used in this study. Section 3 presents the UAV flights and other relevant measurements during the AROMAT campaign. Section 4 presents the methods we used to analyse our measurements. Section 5 presents the SWING-UAV measurements and compares them with coincident measurements, in particular two other DOAS systems.

## 2   The SWING-UAV observation system

### 2.1   The SWING payload

The Small Whiskbroom Imager for atmospheric compositioN monitorinG (SWING) was developed at BIRA-IASB based on the experience gained with previous airborne DOAS instruments (Merlaud et al., 2011, 2012) and in the framework of a collaboration with the "Dunarea de Jos" University of Galati, Romania.

Figure 1 presents a schematic diagram of the SWING instrument, which is displayed with its open housing in Fig. 2. SWING

is based on an Avantes AvaSpec-ULS2048 spectrometer, whose optical bench follows the Czerny-Turner design with a 75 mm focal length and a numerical aperture of 0.07. The entrance slit is 50 μm wide and the groove density of the grating is 600 l/mm. The spectrometer covers the wavelength range 200-750 nm at a spectral resolution of 1.2 nm Full Width at Half Maximum (FWHM). The detector is a SONY ILX554B, which is an uncooled CCD linear array with 2048 pixels of 14x56 $\mu m^2$. SWING uses the standard Avantes AS-5216 microprocessor board with its 16-bit analog-to-digital converter. This board reads a scan

in 1.8 ms and is able to perform on board signal processing such as averaging scans. In SWING, the spectrometer board is powered through USB by a PC-104 (Lippert CSR LX800), located under the spectrometer board. The PC-104 runs the acquisition software and stores the spectra. Both operations are performed on a 2 GB SSD integrated on the PC-104, to avoid the extra weight of an external disc.

    SWING collects the light with a scanning mirror (elliptical Edmund Optics coated with Enhanced aluminium, 12.7 x 17.96

mm). This mirror is mounted on the shaft of a HITEC HS-5056-MG servomotor which is controlled by an Arduino Micro. The latter also interfaces a pressure/temperature sensor (Bosch BMP180) and is linked to the PC-104 via USB. After the scanning mirror, light is collected by a fused silica collimating lens (an Avantes COL-UV/VIS, with a focal length of 8.7 mm and diameter of 6 mm) and through a 26 cm long optical fiber of 400 μm diameter. The mirror is able to scan at ± 55° around the



nadir direction. The instantaneous and angular field of view (FOV) are 2.6° and 110°, respectively. It is also possible to record spectra in the zenith direction by rotating the scanning mirror at 90° relatively to nadir, pointing to a zenith mirror, which is adapted from an Avantes right-angle collimator (COL-90-UV/VIS).

Except the scanner support which is made of aluminium, the structural parts of SWING and its housing are in plastic material (ABS). They were manufactured by 3D printing to optimize their weight and shape. The optical windows are rectangular in nadir and circular in zenith. These windows are in Zeonex, a plastic material with a transmittance above 90% between 350 and 1100 nm.

Table 1 summarizes the main characteristics of SWING. The weight, size, and power consumption of SWING are respectively 920 g, 27x12x8 $cm^3$, and 6 W. The whole system is powered by 5 V, which is supplied by a compact LiPo battery through an UBEC DC-DC converter. More technical details about the electronics circuits, miniaturization efforts, and first tests of SWING on an ultralight trike are presented in Merlaud (2013). The first test flights with SWING onboard the UAV were performed near Galati, Romania, they are presented in Merlaud et al. (2013).

## 2.2 The Flying wing UAV

Figure 3 shows the UAV which was used in this study. It was developed at the "Dunarea de Jos" University of Galati, in parallel with the development of SWING at BIRA-IASB. The aircraft is a flying wing type with a 2.5 m wingspan. It is electrically propelled with LiPo batteries. The total mass of the system, including batteries, is 13 kg. The SWING payload is attached on the back of the wing and powered through its own LiPO battery. During flight, the GPS and attitude angles are recorded at 4 Hz on a SD card. The inertial sensor is a MPU 6050 from Invensense.

The UAV takes off with a catapult and lands on its belly. Airspeed is approximately 100 km.h$^{-1}$ and it has an autonomy of around 1h depending on the flight pattern and wind conditions. The maximum flying altitude is 3 km above ground level. After take-off, the UAV can be piloted from the ground or operated in autopilot mode, following a pre-programmed flight pattern. Combining this with a real-time video link from a camera mounted on the UAV enables to fly the aircraft beyond visual range. Note however that this was not possible during the AROMAT campaign due to a technical issue at the beginning of the campaign (see Sect. 3.2).

## 3 The AROMAT campaign and the SWING-UAV flights

### 3.1 The AROMAT campaign

The Airborne ROmanian Measurements of Aerosols and Trace gases (AROMAT) campaign took place in Romania in September 2014. The AROMAT campaign was supported by the European Space Agency (ESA) in the framework of its Living Planet Programme. The primary objectives of the AROMAT campaign were (i) to test recently developed airborne observation systems dedicated to air quality satellite validation studies such as the AirMAP (Schönhardt et al., 2015), the KNMI $NO_2$ sonde





(Sluis et al., 2010), and SWING, and (ii) to prepare the validation programme of the future Atmospheric Sentinels, starting with Sentinel-5 Precursor.

The AROMAT campaign focused on two locations: the city of Bucharest (44.45° N, 26.1° E), capital and largest city of Romania, and the large power plants of the Jiu Valley, in particular Turceni (44.67° N, 23.41° E, 116 m a.s.l.). Meier et al. (2017) present some of the work done during AROMAT in Bucharest. The SWING-UAV observations, which are the focus of this paper, were only performed in Turceni. Note that an overview presentation of AROMAT and its follow-up AROMAT-2 in August 2015 is the subject of a dedicated paper in preparation.

The coal-fired power station of Turceni is located in a rural area of the Jiu Valley, between the cities of Craiova and Targu Jiu and 210 km west of Bucharest. It is the largest power plant in Romania, producing 1600 MW of electricity. The power plant has four 280m tall smokestacks. The Jiu Valley area and its power plants were chosen as geophysical targets of the AROMAT campaigns since emissions of $NO_2$ and $SO_2$ from this area are clearly visible from space, as was already observed two decades ago by Eisinger and Burrows (1998), and more recently by e.g. Krotkov et al. (2016).

## 3.2 The SWING-UAV flights in Turceni

We performed four SWING-UAV flights, one on 10 September, three on 11 September 2014. The UAV always took off from a flat grass field north of the power plant (44.68° N, 23.40° E, 116 m a.s.l.) and landed there as well. During the first three flights, we performed loops of 1.5 km diameter at around 700 m a.g.l around the take-off site. During these flights, SWING was scanning ± 50° around nadir in steps of 4°, and recording spectra with a total integration time of 400 ms (8x50 ms exposures). Zenith spectra of 2.5 s total integration time (500x5 ms) were acquired every 500 nadir spectra. Figure 4 shows the UAV track of the first flight (performed between 08:40 and 09:25 UTC) of 11 September 2014.

The fourth and last SWING-UAV flight (performed on 11 September 2014 between 14:58 and 15:10 UTC), was dedicated to vertical soundings. For this flight, the scanner was set in zenith position and SWING recorded spectra around zenith (given the varying plane attitude), with a total integration time of 1 s (with exposure times of 5 ms). The UAV made two successive ascents and descents, flying first to 700 m a.g.l., descending to 100 m a.g.l., then doing another ascent to 500 m a.g.l. before landing. This flight pattern was intended to provide information on the absorbers vertical distribution.

Only the first and last flights of 11 September 2014 are analysed in this study. For the sake of clarity, we will hereafter refer to them as the *mapping* and *sounding* flights, respectively.

Note that a first flight was originally planned on 8 September 2014 but failed due to a pilot error at take-off. The UAV and the SWING payload were slightly damaged but could be quickly repaired on site. However, this led us to give-up the use of the autopilot during the flights, for security reasons. Flying with the autopilot would have enabled us to cover a larger area and to adopt more efficient mapping patterns around the whole power plant. It would also have allowed to fly at higher altitude (1500 m a.s.l.), well above the emission plume of the power plant. Without autopilot, all the flights were performed in visual range with a permanent control of the UAV by the pilot on the ground. The flight patterns in this study are consequently not representative of the nominal capabilities of the SWING-UAV observation system.



### 3.3 Coincident measurements

The data analysis of the SWING-UAV experiments benefits from data collected with other instruments operated in Turceni during AROMAT, both in the air and from the ground.

Figure 4 describes the set-up of the AROMAT campaign in Turceni. As described in the previous section, the UAV was operated from the field to the north of the power plant. In the meantime, static ground-based measurements were performed at the Turceni soccer field (44.679° N, 23.378° E, 122 m a.s.l.). In particular, a scanning UV lidar (355 nm) was pointed toward the plume and used to determine the boundary layer height and estimate the aerosol extinction profile, a HORIBA APNA-370 was monitoring the volume mixing ratios of $NO_x$ at the surface and a weather station was recording the meteorological parameters (wind direction and speed, ground temperature and pressure, and relative humidity).

Figure 5 shows two extinction profiles retrieved from the lidar measurements around 9 UTC on 11 September 2014, when we performed the SWING-UAV $NO_2$ experiment. The blind zone of the lidar above ground is 100 m thick. The two extinction profiles exhibit a rather similar shape, with a maximum extinction at 700 m, but the maximum decreases from $5.4 \pm 1 \times 10^{-4}$ to $3.2 \pm 1 \times 10^{-4}$ for 08:30 and 09:30 UTC, respectively.

Several KNMI $NO_2$ sondes (Sluis et al., 2010) were launched on weather balloons on the days of the SWING-UAV observations. The balloons were launched from three different spots: the aforementioned UAV and soccer fields, and a parking lot in the village of Turceni (44.68° N, 23.37° E, 127 m a.s.l., see Fig. 4). We tried several take-off sites for these balloons since it appeared difficult to cross the exhaust plume of the power plant with a wind-driven balloon.

Figure 6 presents the sonde measurements of relative humidity, potential temperature, and $NO_2$ mixing ratio for the two balloon flights at 08:07 and 10:46 UTC on 11 September 2014. Both balloons were launched from the parking lot but, as can be seen from the $NO_2$ profiles, the first one missed the main plume. The measurements at 10:46 UTC indicate that the main plume was then below 800 m, with a maximum detected volume mixing ratio (vmr) of 60 ppb close to this ceiling. The surface vmr is significantly lower (20 ppb), which can be understood from the emission height of the power plant stacks. An elevated $NO_2$ layer is visible on both soundings: at 800 m at 08:07 UTC and at 1200 m at 10:46 UTC.

The sonde meteorological measurements enable to estimate the boundary layer height at the sounding times: around 700 m at 08:07 UTC and around 1100 m at 10:46 UTC. This second altitude is higher than the main plume, which indicates that the latter does not reach the top of the boundary layer. This is understandable since the $NO_2$ field is sampled by the sonde very close to its source. The height and shape of the main $NO_2$ plume appear in relative agreement with the lidar-retrieved extinction profile. Note that the elevated $NO_2$ layer detected on both soundings is just above the boundary layer. The fact that it is also visible during the first sounding, which missed the main plume, and the Mobile-DOAS measurements (see below) indicates that this second plume has a larger horizontal extent than the main plume. This may indicate a remote origin, such as the power plants of Isalnita or Craiova, respectively located 40 and 50 km upwind of Turceni.

Finally, three DOAS instruments were operated during the SWING-UAV $NO_2$ flight on 11 September 2014. The AirMAP (Schönhardt et al., 2015) instrument was mapping the plant exhaust plume from 3 km altitude on board the FUB Cessna, whereas a zenith-sky car-based DOAS system (Constantin et al., 2013) was operated along the roads marked in purple in



Fig. 4. The measurements from these two DOAS instruments are compared with the SWING $NO_2$ dataset in Sect. 5.1. The third DOAS instrument was a car-based system with two viewing directions (zenith and 30° above the horizon) (Merlaud, 2013). This instrument was installed on a car parked in the UAV field in front of the power plant during the SWING-UAV $NO_2$ flight. Interestingly, this instrument missed the main plume during the UAV flight but did detect a small $NO_2$ layer, with a

vertical column density of 6 x $10^{15}$ molec.cm$^{-2}$). This value is consistent with the elevated layer of $NO_2$ seen by both balloons. These static DOAS measurements are used (see Sect. 4.3) to estimate the column in the SWING reference spectrum.

## 4    Data analysis

The data analysis starts with the DOAS analysis of the spectra, which extracts the molecular absorptions along the optical path of the measurements, namely the slant column densities (SCDs). These SCDs are then combined with the UAV position

and attitude information. Regarding $NO_2$, the georeferenced SCDs are converted to vertical column densities (VCDs) by modelling the light path. Regarding $H_2O$, the SCDs recorded at different altitudes yield the volume mixing ratio of water vapour in between these altitudes.

### 4.1    Spectral analysis

The DOAS analysis is a well-established technique (Platt and Stutz, 2008) to retrieve molecular absorptions in the UV-Vis

range. It is based on the fact that the absorption cross-sections of certain molecules, such as $NO_2$, $O_3$, or $H_2O$, vary much more rapidly with wavelength than Rayleigh and Mie scattering. In practice, the spectrum to analyse is divided by a reference spectrum to remove solar Fraunhofer structures and reduce instrumental effects. The components in the logarithm of this ratio (the measured optical depth) which vary slowly with the wavelength are filtered out by a low-order polynomial while the remaining high frequency structures are simultaneously fitted with high-pass filtered laboratory cross-sections. The DOAS

analysis of a spectrum yields the integrated concentration of an absorber along the optical path of the measurement, with respect to the same quantity in the reference spectrum. This quantity is called the differential slant column density (DSCD).

Table 2 lists the DOAS analysis settings used in this study to retrieve $NO_2$ and $H_2O$ DSCDs. We used the QDOAS software, which was developed at BIRA-IASB (Fayt et al., 2011; Danckaert et al., 2017), to analyse the spectra. In addition to interfering species ($O_3$ and $O_4$), we fitted the so-called Ring contribution (Grainger and Ring, 1962), which corresponds to the filling-in of

the solar Fraunhofer lines caused by rotational Raman scattering on $O_2$ and $N_2$. For both species, we also fit an intensity offset and correct for small changes in spectral resolution during the flight. The latter is achieved by fitting a synthetic cross section corresponding to the derivative of the solar reference with respect to the slit function width (Beirle et al., 2017; Danckaert et al., 2017). Note that for the $H_2O$ fit, our DOAS settings are inspired by Wagner et al. (2013). As mentioned in Sect. 3.2, the scanner periodically took zenith spectra with a longer integration time (5 s). The reference spectra were chosen amongst

these zenith measurements to minimize the absorber residual column, i.e. outside the $NO_2$ plume and at the maximum altitude reached with the UAV.



Figure 7 shows typical DOAS fits for $NO_2$ (left) and $H_2O$ (right) for spectra recorded from the UAV during the AROMAT campaign, together with the fitted $O_4$ and the fit residuals. The ratio of the fitted DSCD to its uncertainty (an output of the DOAS analysis) yields an estimate of the signal-to-noise ratio of the DSCD measurements. It is around 18 for the $NO_2$ DSCD inside the plume and 30 for the $H_2O$ DSCD. Note that the uncertainties also indicate the 1-$\sigma$ DSCD detection limit for the two species, i.e. 3 x $10^{15}$ molec.cm$^{-2}$ and 1.4 x $10^{21}$ molec.cm$^{-2}$ for $NO_2$ and $H_2O$, respectively.

Figure 8 presents $NO_2$ DSCD time series corresponding to the mapping flight on 11 September 2014. The UAV was performing loops at 700 m a.g.l, which are shown in Fig. 4. The $NO_2$ DSCD times series indicates that the SWING line-of-sight crossed the plume at each loop, with a periodic pattern showing $NO_2$ DSCD peaks between 0.06 and 1.3 x $10^{17}$ molec.cm$^{-2}$. This indicates that the UAV was flying at the border of the $NO_2$ plume, as is further discussed in Sect. 5.1.

Figure 14 presents (upper panel) the $H_2O$ DSCD measurements corresponding to the sounding flight on 11 September 2014. As described in Sect. 3.2, the SWING scanner was set for zenith measurements during the entire flight. The two broad structures in the $H_2O$ DSCD correspond to the two successive ascents, the second one reaching a lower altitude. These $H_2O$ measurements are further discussed in Sect. 5.2.

## 4.2 Georeferencing

As the SWING payload contains neither a GPS nor an attitude sensor, the SWING PC was synchronized to the GPS time before each flight. The georeferencing of the measurements was performed in post processing using the UAV GPS and attitude data recorded on the SD card during the flight (see Sect. 2.2).

Figure 9 presents an excerpt of the time series of the attitude angles recorded by the UAV autopilot during the first flight on 11 September 2014. The excerpt corresponds to one of the UAV loops shown in Fig. 4. The roll lies between 30° and 60° and is mainly negative, indicating that the UAV keeps rotating in the same direction. The pitch angle lies between -12° and 30°. Overall, the UAV attitude appears to vary a lot when compared to a manned aircraft flying at higher altitude. This is partly due to the turbulences in the boundary layer and would be reduced if flying in autopilot.

The attitude angles were used together with the UAV GPS measurements and the SWING scanner positions to georeference the SWING-UAV spectra and DSCDs using the geometric formulas given by Schönhardt et al. (2015).

## 4.3 Vertical columns of $NO_2$

As described in Sect. 4.1, the DSCDs retrieved with the DOAS analysis not only depend on the absorber concentration, but also 1) on the residual column in the reference spectrum and 2) on the light paths of the measurements. It is thus necessary to address both issues to get a more practical geophysical quantity, namely the vertical column density (VCD), i.e. the absorber concentrations integrated along the vertical path.

The reference spectrum for $NO_2$ was chosen among the SWING zenith spectra recorded at a high altitude and when the UAV was outside the main plume. The latter criterion could be checked with the coincident Mobile-DOAS measurements. However, as can be derived from the balloon-borne $NO_2$ sonde measurements (see Fig. 6), a second layer of $NO_2$ was present above the main plume, which covered a larger area. The absorption of this second layer is present in our reference spectrum and needs





to be accounted for. The two sonde measurements lead to two different VCD values for this elevated $NO_2$ layer, respectively 6.2 x $10^{15}$ molec.cm$^{-2}$ at 08:07 UTC, and 1.2 x $10^{16}$ molec.cm$^{-2}$ for the sonde at 10:46 UTC. None of these sondes was simultaneous with the SWING-UAV flight. However, the $NO_2$ column of the elevated layer was measured during the UAV flight by the double channel Mobile-DOAS, being around 6 x $10^{15}$ moles.cm$^{-2}$. Considering that the reference spectrum

was recorded in zenith direction and assuming a geometric AMF of 1, this column value was used as the reference column ($SCD_{ref}$) to convert the DSCDs to SCDs according to:

$$SCD = DSCD + SCD_{ref} \qquad (1)$$

The light paths are accounted for in this study by estimating an air mass factor (AMF) for each spectrum. AMFs were calculated using the radiative transfer model UVspec/DISORT (Mayer and Kylling, 2005). The latter is based on the discrete ordi-

nate method and deals with multiple scattering in the pseudo-spherical approximation. The ancillary measurements described in Sect. 3.3 were used as geophysical inputs: the profiles of $NO_2$ and aerosol extinction were taken from the sonde (see Fig. 6) and lidar measurements (see Fig. 5), respectively. Meier et al. (2017) have developed a method to estimate the ground albedo from the uncalibrated radiances recorded by the AirMAP instrument. The ground albedo was not retrieved above Turceni but we used the albedo value retrieved from AirMAP above a surface of similar type (grass) in the Bucharest flight of 8 September

2014, where the albedo was 0.02. The typical AMF for SWING-UAV nadir observations during AROMAT is 2.5. The VCD is then expressed as:

$$VCD = SCD/AMF \qquad (2)$$

The error on the retrieved VCD depends on the uncertainties on the elements of Eq. 2.

Note first that considering the short time span of the SWING-UAV flights (40 minutes), the stratospheric content of $NO_2$ is

assumed to be constant. Its contribution to the $NO_2$ column thus cancels in the DSCD fit.

Regarding the reference $NO_2$ column ($SCD_{ref}$), it was quantified by coincident Mobile-DOAS measurements. Its uncertainty is thus also neglected.

The uncertainties on the retrieved VCD originate then from two sources: the DSCD and the AMF.

The random error on the DSCD, $\sigma_{DSCD}$, is estimated during the DOAS analysis from the fit residuals, this is a standard

output of the QDOAS software. We used this standard output as the uncertainty on our DSCD. Systematic uncertainties in the DSCD are mainly related to spectroscopy and are neglected in this study.

The error on the AMF depends on the errors on the radiative transfer model inputs with respect to the true state. Previous studies indicate that the two main uncertainty sources for nadir observations of the $NO_2$ column are the ground albedo and the relative position of the $NO_2$ and aerosol layer (Leitão et al., 2010; Meier et al., 2017; Tack et al., 2017). In our case, the lidar

and sonde indicate that the $NO_2$ and aerosols have a similar profile, as can be expected considering the large source in a rural area. On the other hand, as mentioned above, the ground albedo was estimated from albedo retrievals at another site three days before. The uncertainty on this parameter is therefore relatively large and we estimate it to 100% of our value (i.e. ± 0.02).





Figure 10 (upper panel) shows air mass factors calculated for flight altitudes of 0.8 km and 3 km, in nadir geometry, and varying the albedo between 0.01 and 0.3. We took the aerosol profile and conditions of the AROMAT dataset as described above. The lower panel of Fig. 10 shows the box air mass factors for two albedos (0.02 and 0.2) for an altitude of 0.8 km (left) and 3 km (right), between the surface and 4 km altitude.

The AMF is larger for the low altitude simulations (corresponding to SWING-UAV observations) when compared to higher altitude (AirMAP observations). This is understandable from the input $NO_2$ profile and the box air mass factors: they indicate that in both cases, the maximum sensitivity is right under the observation altitude. For the SWING-UAV observations, this means a maximum sensitivity inside the plume and thus a larger AMF when compared to AirMAP. Interestingly, the albedo dependency is also less pronounced for the low altitude simulations. In both cases, reducing the albedo will increase the fraction

of the collected photons which are back scattered in the atmosphere before hitting the ground, and thus decreases the AMF. The difference lies in the fact that for the high altitude case, photons can be scattered above the plume whereas for the low altitude one, the scattering takes place within the plume anyway. In Fig. 10 (right lower panel), the box air mass factor remains above 2 between 1 and 2.5 km altitude for the low albedo scenario, indicating a high sensitivity in this clean layers even for a low albedo.

Considering the uncertainty on the albedo discussed above (0.02), the AMF simulations were used to estimate an associated error ($\sigma_{AMF}$) of 0.2 on the AMF for the SWING retrievals. In practice, we consider a relative uncertainty of 10% (ca. 0.25 in absolute values) on the AMF, to take into account other minor error sources.

The errors are then summed in quadrature to get the total error on the $NO_2$ VCD:

$$\sigma_{VCD_{tropo}} = \sqrt{\left(\frac{\sigma_{DSCD}}{AMF}\right)^2 + \sigma^2_{AMF}\left(\frac{DSCD}{AMF^2}\right)^2} \tag{3}$$

These uncertainties are displayed as error bars in Fig. 13. They typically range between 2 and 4 x $10^{15}$ molec.cm$^{-2}$.

## 4.4   Volume mixing ratio of $H_2O$

During the sounding flight (see Sect. 3.2), the scanner was fixed in zenith position. This flight was dedicated to sound the atmosphere vertically by performing two successive ascents and descents. Unfortunately, this flight missed most of the $NO_2$ plume but it appears possible to retrieve information on the water vapour abundance in the boundary layer.

Consider two zenith DSCD measurements of an absorber $a$ recorded at different altitudes ($A$ and $B$). Assuming that the light scattering is identical for the two measurements and that this scattering takes place above the highest of these two altitudes, the difference between the zenith DSCDs directly yields the partial VCD ($VCD_{AB}$) between the two altitudes $A$ and $B$:

$$VCD_{AB} = DSCD_A - DSCD_B \tag{4}$$





This partial vertical column is by definition the concentration integrated vertically between $A$ and $B$. Assuming air to be an ideal gas, this partial column can be expressed as:

$$VCD_{AB} = \int_A^B C_a \frac{pN_a}{RT} dz \qquad (5)$$

where $C_a$ is the volume mixing ratio of the absorber $a$, $p$ the air pressure, $z$ the altitude, $N_a$ the Avogadro number, $R$ the ideal gas constant, and $T$ the air temperature.

Assuming a constant temperature and volume mixing ratio between $A$ and $B$, and expressing the air pressure with the isothermal scale height ($z_0 = RT/\mu g$, where $\mu$ is the molar mass of air and $g$ the acceleration of gravity), Eq. 5 leads to:

$$VCD_{AB} = \frac{C_a \Delta P_{AB} N_a}{g\mu} \qquad (6)$$

where $\Delta P_{AB}$ is the difference in pressure between the two altitudes $A$ and $B$. It then follows the expression of $C_a$, the volume mixing ratio of $a$:

$$C_a = \frac{g\mu VCD_{AB}}{\Delta P_{AB} N_a} \qquad (7)$$

The uncertainty on $C_a$ is obtained by propagating the errors on $VCD_{AB}$. From Eq. 4 and assuming a similar uncertainty ($\sigma_{DSCD}$) on the DSCDs for $A$ and $B$, the error on the volume mixing ratio $\sigma_{C_A}$ is written as:

$$\sigma_{C_A} = \sqrt{2}\sigma_{DSCD} \frac{g\mu}{\Delta P_{AB} N_a} \qquad (8)$$

Note that the above formula neglects the error on the measured pressure difference $\Delta P_{AB}$. From Eq. 7, we also assume that the molar mass of air ($\mu$) is independent of the volume mixing ratio of the investigated absorber ($C_a$), which is wrong in principle. However, even for a large mixing ratio of water vapour in the boundary layer, the associated error is negligible when compared to the effect of the uncertainty on the DSCD. Note also that close to a large aerosol source like the Turceni power plant, the largest part of the uncertainty on $C_a$ may originate from our very fist assumption, i.e. the assumed constant scattering conditions between the two altitudes. This could be wrong especially if the plane is below or inside the main exhaust plume for one of the two points.

The above equations are used hereafter (Sect. 5.2) to estimate the volume mixing ratio of water vapour in the boundary layer. Note that the same technique may be used to retrieve information on the abundance and profile of other absorbers.





## 5 SWING-UAV measurements around the Turceni power plant

### 5.1 The horizontal distribution of $NO_2$ around the power plant

Figure 11 presents two maps of the $NO_2$ VCDs measured on 11 September 2014 around the Turceni power plant. The upper panel shows the AirMAP and Mobile-DOAS measurements whereas the lower panel shows the SWING measurements. The plume extent appears clearly in the AirMAP data: originating from the power plant, the plume is pushed westwards by the wind. The width of the plume quickly increases to reach 2 km above the Turceni village, ca 3 km downwind of the plant. The length of the plume is above 12 km but it was not fully sampled by AirMAP. As expected, the horizontal gradients of the $NO_2$ VCD field are steep, ranging from the background (around 2 x $10^{15}$ molec.cm$^{-2}$) to the plume (2 x $10^{16}$ molec.cm$^{-2}$) in a few hundred meters. The plume itself is not homogeneous but includes areas with $NO_2$ VCDs above 2.5 x $10^{16}$ molec.cm$^{-2}$, not only close to the source, but along its entire observed length.

The SWING map on the lower panel of Fig. 11 covers a much smaller area than AirMAP, a circle with a diameter of approximately 3 km. Nevertheless, it shows the edge of the plume, which also enables to observe the sharp horizontal gradients of the $NO_2$ field between the plume and the background. The absolute values of the $NO_2$ VCDs retrieved from SWING appear similar to the ones retrieved from AirMAP. Note that the dashed white lines on the SWING map correspond to the edges of the plume as seen by AirMAP.

Figure 12 presents a scatter plot to compare quantitatively the AirMAP and SWING $NO_2$ VCDs. The AirMAP measurements taken into account for this comparison were recorded in two adjacent flight lines, between 08:40 and 08:41 and between 08:48 and 08:50. These AirMAP data are almost perfectly time-coincident with the SWING-UAV measurements, which started at 08:40 and ended at 09:25. The AirMAP-SWING pairs of points were built in two steps. First, we applied a set of AMFs to the AirMAP slant columns. These AMFs are consistent with the ones used for SWING (see Sect. 4.3), adapting the altitudes and geometry to the AirMAP conditions but using the same ground albedo (0.02), $NO_2$, and aerosol extinction profiles. Secondly, all AirMAP VCDs within 100 m of a SWING pixel were averaged to produce one averaged AirMAP point. AirMAP and SWING $NO_2$ VCDs are up to 3.8 and 4.1 x $10^{16}$ molec.cm$^{-2}$, respectively. The Pearson correlation coefficient between the two dataset is 0.88, while the slope is 0.91, the AirMAP VCDs being slightly below the SWING ones. This higher values for the SWING VCDs with respect to AirMAP can also be seen in Fig. 13.

We have also compared the two airborne DOAS products with time coincident car-based DOAS measurements. As mentioned in Sect. 3.3, the UGAL zenith-only Mobile-DOAS instrument was also operated during the SWING-UAV flight, along the road between the power plant and the Turceni Village.

Figure 13 presents the $NO_2$ VCDs measurements by the three DOAS instruments along the road driven by the Mobile-DOAS (see Fig. 11) with respect to the longitude. The Mobile-DOAS did several drives, back and forth, along the road, but the main visible trends are stable. From East to West, the $NO_2$ VCD increases as we enter the plume. This is visible for the three instruments and according to the Mobile-DOAS drives, the edge of the plume seems rather static at 23.395° E. However, the Mobile-DOAS data also show a dip in the $NO_2$ VCDs at 23.382° E. This dip corresponds to the right angle turn close to the bottom of the red area in the upper panel of Fig. 11. It is close to the southern edge of the plume. Although this pattern appears



to be reproducible in the three Mobile-DOAS drives of Fig. 13, it is not seen in the airborne data. Note that, both before and after the dip, the Mobile-DOAS VCDs are significantly higher (by approximately a factor of 2) than the airborne VCDs, but that this scaling factor is reduced in the western part, i.e. when the Mobile-DOAS was well inside the plume.

The differences observed between the two airborne DOAS systems may originate from the small time difference, which could be linked with variations in the power plant $NO_2$ emissions. It could also be caused by errors on the radiative model inputs (in particular since a constant ground albedo is assumed). The comparison with the Mobile-DOAS also suggests another effect to take into account: the limitations of our 1D radiative transfer model to take into account the complex 3D geometry of the exhaust plume. Indeed, the airborne VCDs on Fig. 13 appear smoothed when compared to the Mobile-DOAS VCDs, which is understandable from the different light paths of the two types of measurements. This motivates further investigation, both on practical intercomparison exercises and on theoretical work to estimate the optimal spatial resolution reachable with an airborne DOAS instrument, e.g. with 3D radiative transfer models such as McArtim (Deutschmann et al., 2011).

## 5.2 Water vapour mixing ratio and relative humidity close to the ground

Figure 14 presents the time series of $H_2O$ DSCDs, pressure, and viewing zenith angle (VZA) as measured during the sounding flight on 11 September 2014 between 14:59 and 15:10 UTC. As described in Sect. 3.2, the SWING scanning mirror was fixed in upward position and the UAV performed two successive ascents, first to 700 m then to 500 m a.g.l.

The $H_2O$ DSCDs signal can be divided in two components, which are illustrated in the upper and lower panel of Fig. 14, respectively. First, there is a slow component visible in the upper panel, with two cycles during the flight. These changes are correlated with the variations in pressure, and thus in altitude. This slow part corresponds to the change of the remaining column of $H_2O$ above the UAV when its altitude changes. Secondly, the higher frequency variations in the DSCD time series are related to the changes of the attitude of the UAV, and thus to changes in VZA. At first order, given an altitude and assuming a homogeneous $H_2O$ field, the slant column increases with VZA around its minimum value when the light path is vertical. This correlation is visible and quantified in the two time series extracted from the descents of the UAV in the lower panels of Fig. 14.

We retrieved the volume mixing ratio of $H_2O$ as presented in Sect. 4.4. In practice, we selected two pairs of points during the flights where both the pitch and roll were under $2°$, and close to the ceiling and bottom of the flight, but above the surface to avoid the inhomogeneities close to the ground. These points are marked in the figure with blue vertical lines. Applying the formulas described in Sect. 4.4 to the two soundings leads to volume mixing ratios of $0.013 \pm 0.002$ and $0.012 \pm 0.002$ mol.mol$^{-1}$, respectively. Converting this value to a relative humidity yields, close to the surface, 41% and 39%, respectively. There were no time coincident measurements of the relative humidity, but the balloon-borne measurements (see Fig. 6) show that it varied between 45% and 70% in the boundary layer during the balloon flight at 10:46. These balloon observations are consistent with the surface measurements of the INOE weather station at the soccer field, which ran until 13:34, when the relative humidity was 38.8%. Our estimated humidities in the boundary layer appear therefore realistic. It is worth noting that this experiment should be repeated with time coincident ancillary observations, but in other places. Indeed, the edge of a large





exhaust plume of a power plant is clearly not optimal for the assumptions of homogeneity and simple scattering required for the retrieval.

## 6 Conclusions

The SWING instrument has been tested from a UAV during the AROMAT campaign in Romania, in September 2014. The SWING-UAV flights were performed in the vicinity of the Turceni power plant. Using the DOAS technique, the $NO_2$ and $H_2O$ abundances were retrieved from the SWING spectra, and used to infer the horizontal distribution of $NO_2$ VCDs and the volume mixing ratio of the water vapour in the boundary layer.

The retrieved SWING-UAV $NO_2$ VCDs are up to 4.7 $\pm$ 0.4 x $10^{16}$ molec.cm$^{-2}$ in the exhaust plume. They agree within 10% with a time coincident airborne DOAS experiment, namely the AirMAP instrument, which was operated from a higher altitude. The SWING-UAV $NO_2$ measurements were also compared with simultaneous Mobile-DOAS measurements from a car. The comparison is less good at the edge of the plume, which may be related to the effective spatial resolution achievable from a plane.

Regarding the $H_2O$ measurements, we retrieved a water vapour mixing ratio of 0.013 $\pm$ 0.002 mol.mol$^{-1}$ in the boundary layer. This finding could not be validated with time-coincident measurements but it is consistent with observations performed a few hours before at the same location.

The UAV was not operated in nominal mode during the AROMAT campaign, due to a technical problem. Overall, it appears an interesting and promising platform to study a local source in the countryside such as a power plant. It is however limited in the areas it can cover, both because of the flight regulations, and because of the flight autonomy.

To further investigate the SWING performances, the instrument was installed on a manned aircraft alongside the AirMAP instrument during the AROMAT-2 campaign. This will be described in a future study.

*Acknowledgements.* The AROMAT activity was supported by ESA (contract $4000113511/15/NL/FF/gp$) and by the Belgian Space Policy (contract $BR/121/PI/UAV\ Reunion$). Regarding the AirMAP instrument, financial support through the University of Bremen Institutional Strategy Measure M8 in the framework of the DFG Excellence Initiative is gratefully acknowledged. We thank the people of Turceni and the Air Traffic Control of Romania for their support and cooperation.





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




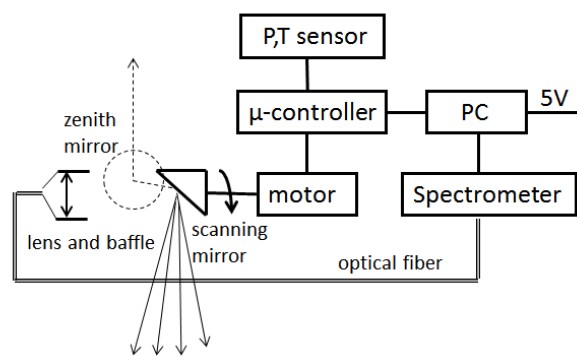

**Figure 1.** Schematic diagram of the SWING instrument.

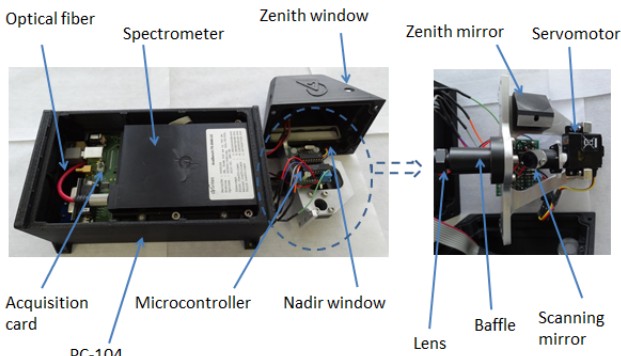

**Figure 2.** The SWING instrument. Its dimensions are 27x12x8 cm$^3$ and its weight is 920 g.

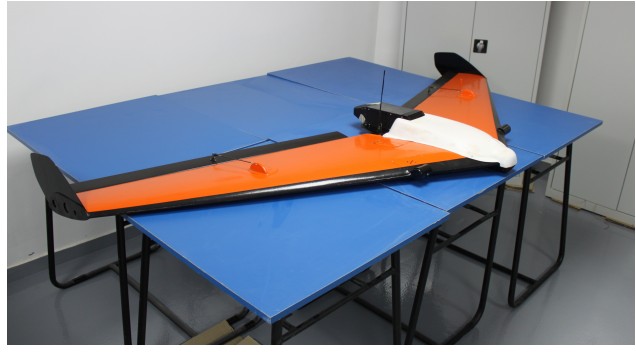

**Figure 3.** The 2.5 m wingspan flying wing UAV developed in parallel with SWING.





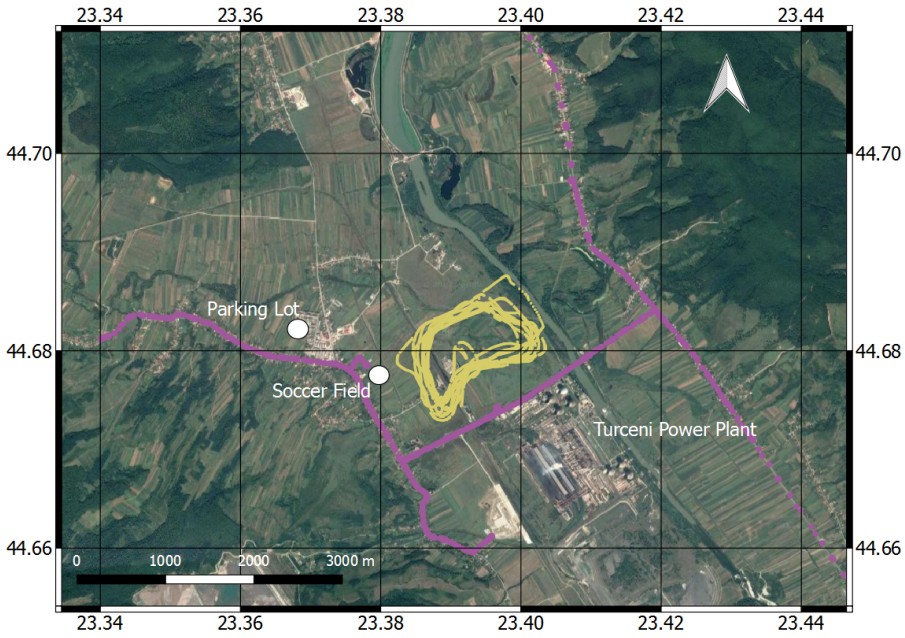

**Figure 4.** The set-up in Turceni during the golden day. The yellow loops indicate the UAV tracks during the first flight on 11 September 2014 while the pink line shows the roads simultaneously used by the Mobile-DOAS.

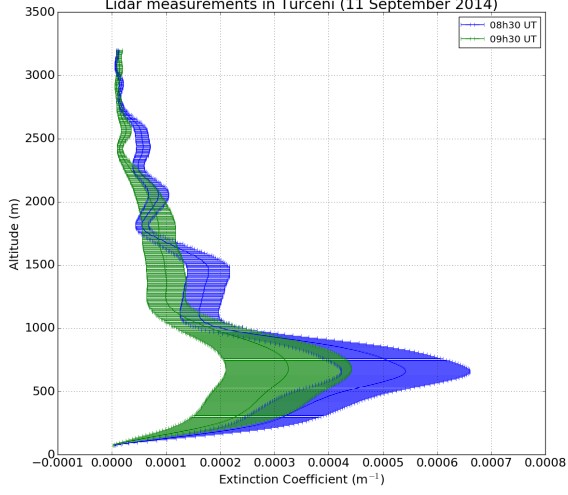

**Figure 5.** Lidar measurements in Turceni on 11 September 2014, at 08:30 and 09:30 UTC.





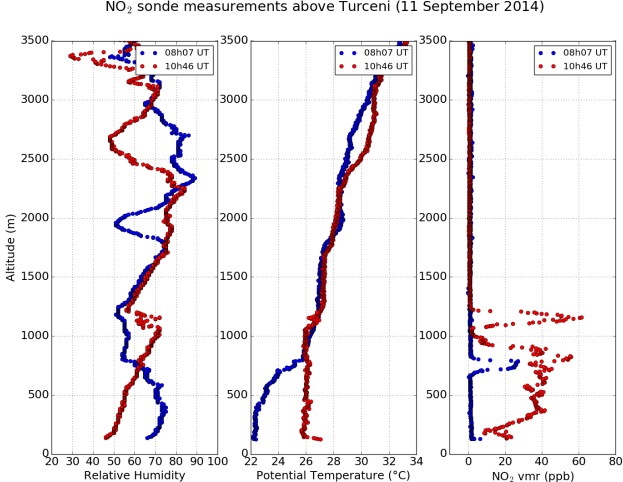

**Figure 6.** Lower parts of two $NO_2$ balloon-borne sonde measurements in Turceni on 11 September 2014. The left panel presents the profiles of relative humidity during the two soundings, the middle panel presents the potential temperature profiles, and the right panel presents the $NO_2$ profiles.

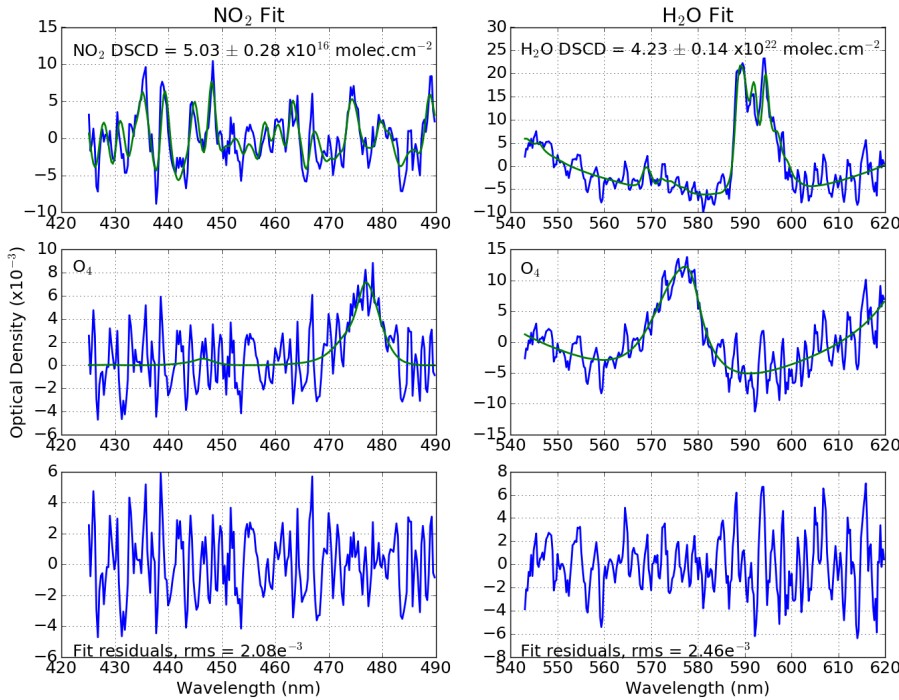

**Figure 7.** Example of DOAS fits of $NO_2$ and $H_2O$ in SWING spectra (upper panels). The figure also shows $O_4$ fits (middle panels) and the fit residuals (lower panels).




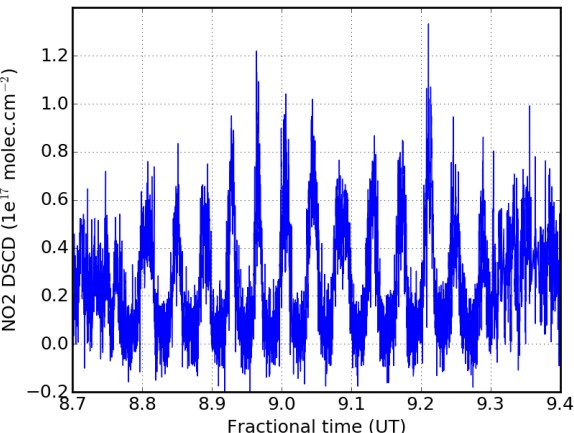

**Figure 8.** Time series of the NO$_2$ DSCDs fitted in the first flight on 11 September 2014.

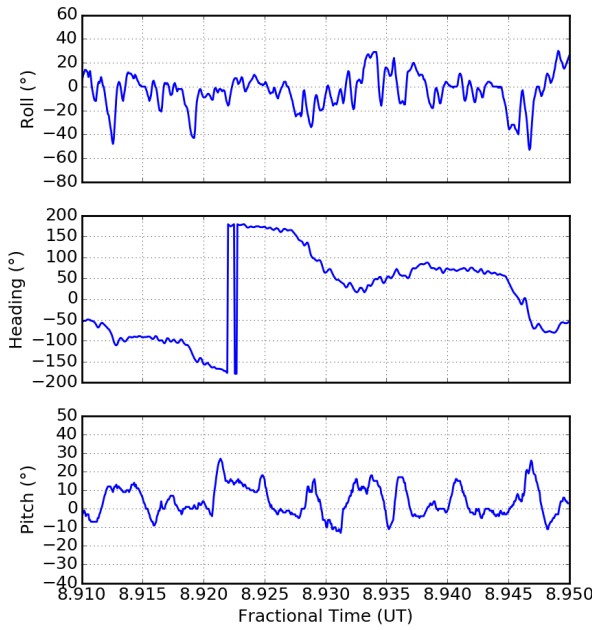

**Figure 9.** Attitude angles measured by the UAV autopilot during one loop of the first flight on 11 September 2014.





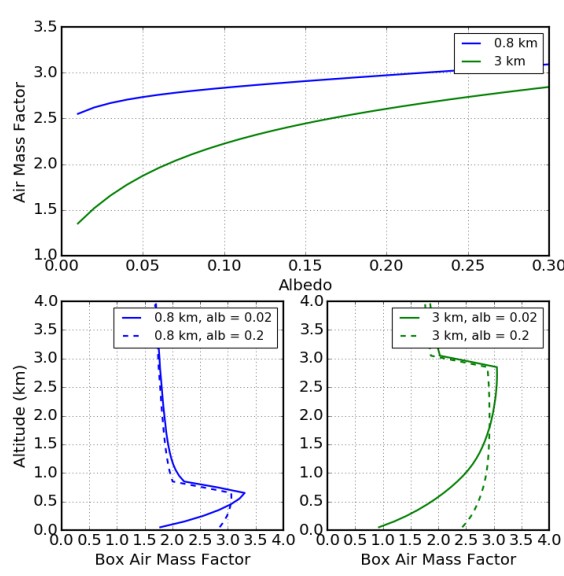

**Figure 10.** Calculated nadir air mass factors in the AROMAT conditions, respectively from 800 m altitude and 3 km altitude.





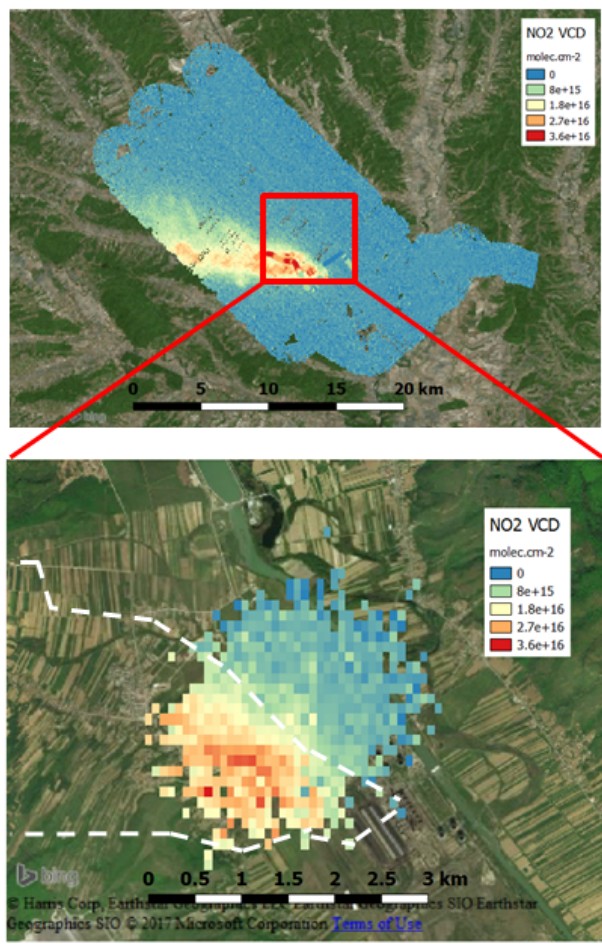

**Figure 11.** SWING, AirMAP, and Mobile DOAS measurements of $NO_2$ in Turceni on 11 September 2014. The upper panel shows the AirMAP and Mobile-DOAS measurements (see text) while the lower panel shows the SWING measurements superimposed on the trace of the plume seen by AirMAP (dashed white line).



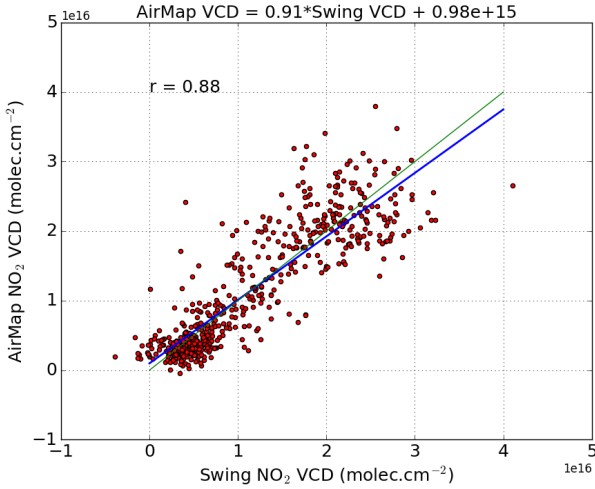

**Figure 12.** Scatter plot of collocated SWING and AirMAP $NO_2$ vertical columns.

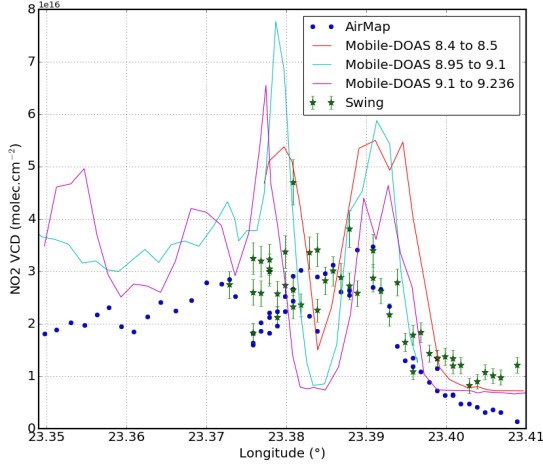

**Figure 13.** SWING, AirMAP, and Mobile DOAS $NO_2$ VCD measurements along the Mobile-DOAS track on 11 September 2014.



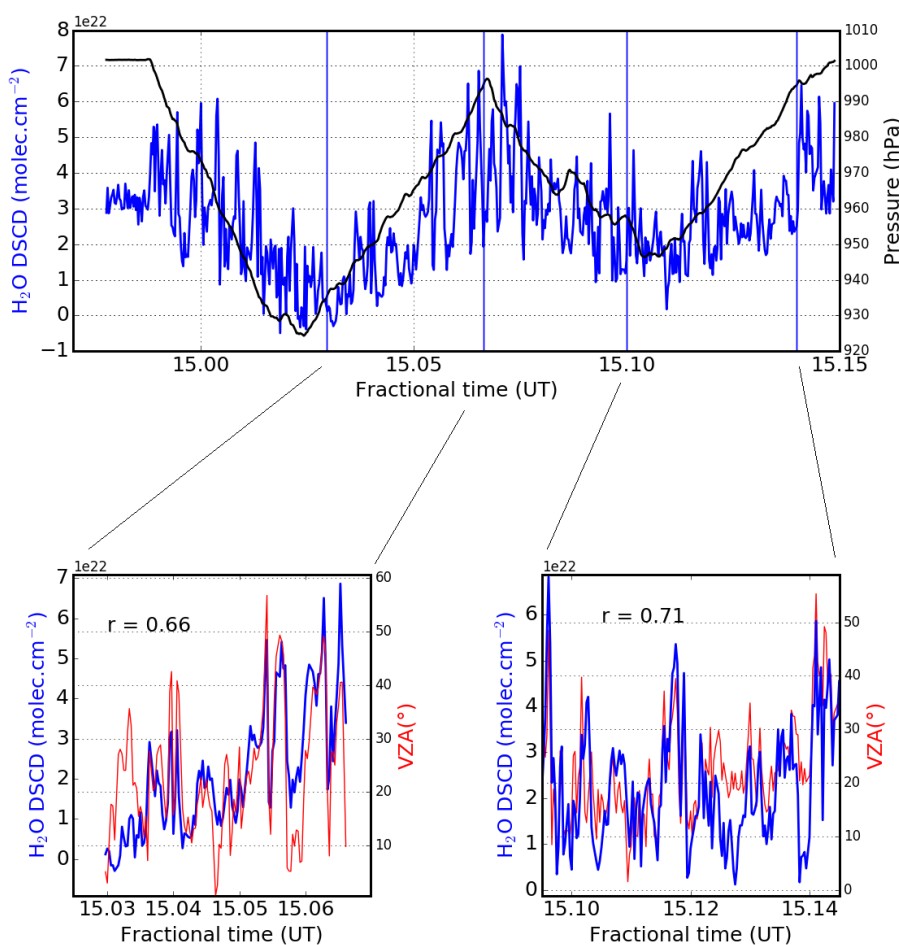

**Figure 14.** Time series of $H_2O$ DSCDs and pressure (upper panel), and of $H_2O$ DSCDs and VZA angles during the descents (lower panel) on 11 September 2014.



**Table 1.** Main characteristics of the SWING instrument.

| | |
|---|---|
| Spectrometer | Avantes AvaSpec-ULS2048 |
| Spectral range | 200-750 nm |
| Spectral resolution | 1.2 nm |
| Instantaneous Field of View | 2.6 ° |
| Pixel size (at 700 m) | 60 m |
| Dimensions | 27x12x8 cm$^3$ |
| Weight | 920 g |
| Power Consumption | 6 W |

**Table 2.** DOAS analysis settings for the $NO_2$ and $H_2O$ fits.

| | $NO_2$ | $H_2O$ |
|---|---|---|
| Fitting window | 431-495 nm | 543-620 nm |
| $NO_2$ (294 K) | Vandaele (1998) | n/a |
| $O_4$ (293 K) | Thalman and Volkamer (2013) | Thalman and Volkamer (2013) |
| $H_2O$ (296 K) | Rothman et al. (2010) | Rothman et al. (2010) |
| $O_3$ (223 K) | Bogumil et al. (2003) | Bogumil et al. (2003) |
| Ring | Chance and Spurr (1997) | Chance and Spurr (1997) |
| Polynomial order | 3 | 5 |
| Small resolution changes | x | x |
| Intensity offset order | 2 | 2 |