# Peer review of "The Small Whiskbroom Imager for atmospheric compositioN monitorinG (SWING) and its operations from an Unmanned Aerial Vehicle (UAV) during the AROMAT campaign"

_Atmospheric Measurement Techniques, 2017_

## Referee Comment (RC1) · Anonymous Referee #1 · 22 Nov 2017

The manuscript by Merlaud et al. describes the retrieval of NO2 and H2O near a power plant from a small unmanned aerial vehicle. The paper describes in detail the instrument set up and data analysis procedures. It fits very well within the scope of AMT.

General comments:

The manuscript details very well the instrument, data analysis and its application during the AROMAT campaign, which makes it a great resource for people working with

similar instruments. Measurement capability and errors are well described, however the conclusions miss a discussion on suitability of the SWING payload as air quality monitoring tool. Though it is obvious that losing the auto pilot option presented a major shortcoming for the AROMAT mission, the manuscript would strongly benefit from a discussion on what has been learned from this campaign on the suitability for more regular monitoring missions and what are the points that are being addressed in the next campaign. Please also explain how the H2O measurements fit into this greater picture.

Overall this paper is well suited for AMT and I recommend publication after extension of the conclusion section.

Specific comments:

Abstract: please include the purpose of the AROMAT campaign, e.g. testing payload feasibility

p.2., line 16ff: paragraph could be shortened, since focus is on unmanned vehicles. Since references are not complete, I recommend using "e.g" for citations.

p.2. line 32, please consider citing a newer reference

Figure 4: please define or omit "golden day"

Technical corrections:

P1., line 14 and ff: "molec.cm-2" Is this properAMT style or should it be molec. x cm-2?

p.2, line 4: "wingspan wider than"

p.5., line 10: "280m" is missing space

p.7, line 5: remove ")"

AMTD

---

## Referee Comment (RC2) · Anonymous Referee #2 · 24 Nov 2017

The paper introduced a compact remote sensing instrument (SWING) dedicated to mapping trace gases from an UAV platform. SWING-UAV experiments were performed in Romania on 11 September 2014 during the AROMAT campaign. Also some simultaneous experiments using airborne imaging DOAS (AirMAP), ground-based DOAS, Lidar and balloon-borne in-situ observations were carried out to evaluate SWING's performances. The reasonable results were obtained from these experiments, i.e. NO2 VCD distributions, H2O volume mixing ratio etc. The SWING instrument could be used to supervise the emission sources and mapping atmospheric pollution in the future. I

consider it to be suitable for publication on AMT after minor corrections as following. 1. The important issue of VCD calculations is the AMFs. During the flight, the AMFs for SWING also depend on viewing angle, geometric angle, i.e. roll and pitch etc. Are these factors considered in the calculation of AMF? 2. The saturated absorption effect of water vapor may bring errors to the spectral retrieve of water vapor. Do the authors consider about this? 3. SO2 is also another main product of exhaust plume from power plant. Since the spectral range of 200-750nm, it would be interesting to have a try to retrieve SO2 ? 4. The middle panel in Figure 6 presents the potential temperature profiles. However, the data show the temperature increases with height. Is it thermal inversion? Inversion is easy to accumulate pollutants. What's the impact on the distribution of emissions of NO2? 5. The Mobile DOAS VCDs are significantly higher (by approximately a factor of 2) than the airborne VCDs. Why? Please give some discussions. 6. Is there any temperature control for the instrument (spectrometer)? For the ascent and descent, how about the spectral stability? 7. The spectral resolution of SWING is 1.2nm or 1.3nm? 1.2nm in table 1, but 1.3nm on Page 3, Line 22.

---

## Author Comment (AC1) · 9 Dec 2017

*We first thank the referee for his review and for his constructive remarks, which were used to improve the manuscript. Our replies to the referee comments are in italic, the changes in the manuscript are in bold.*

Measurement capability and errors are well described, however the conclusions miss a discussion on suitability of the SWING payload as air quality monitoring tool. Though it is obvious that losing the auto pilot option presented a major shortcoming for the AROMAT mission, the manuscript would strongly benefit from a discussion on what has been learned from this campaign on the suitability for more regular monitoring missions and what are the points that are being addressed in the next campaign. Please also explain how the H2O measurements fit into this greater picture.

*We have expanded the conclusion to take into account the referee comment, which addresses the different points mentioned by the referee. A dedicated paper is in preparation for the overview of what was learnt during the AROMAT-1 and AROMAT-2 campaigns.*

*Regarding the H2O vmr measurements, the result came out of the sounding flight but the latter was initially dedicated to NO2. It appeared that, for the UAV and for the balloons, so close to the source, it was actually difficult to fly well inside the narrow plume, so we could not get the NO2 vmr from the spectra. Nevertheless, we consider it worth to do the analysis for H2O since the same scheme can be applied to NO2 in a place with a more homogeneous NO2 surface layer. Beside that, knowledge of relative humidity is also useful to estimate the optical properties of the aerosol affected by hygroscopic growth, for instance with the widely used OPAC aerosol model. We added a sentence mentioning this with the reference in section 4.4:*

**This information could be used to estimate the optical properties of aerosol growing with increasing relative humidity, using for instance the OPAC model Hess(1998).**

*-----*

Specific comments:

Abstract: please include the purpose of the AROMAT campaign, e.g. testing payload Feasibility

*We have modified the abstract adding in the second paragraph after AROMAT*

**which was dedicated to test newly developed instruments in the context of air quality satellite validation**

p.2., line 16ff: paragraph could be shortened, since focus is on unmanned vehicles

*It is true that we focus on UAV but we also present measurements from a manned aircraft in the paper (AirMAP from the FUB Cessna). Moreover, there are not many DOAS studies from UAVs yet and our work builds at least as much on previous DOAS experiments from traditional aircraft than on other UAV atmospheric experiments cited in the first paragraph, which are mainly in situ experiments. Therefore, we prefer to keep these manned aircraft references.*

Since references are not complete, I recommend using "e.g" for citations.

*We agree with this remark and have inserted "e.g." in the first and second paragraph at the five places where a list of studies was used to give examples.*

p.2. line 32, please consider citing a newer reference

*We agree that the reference from 2006 does not appear wise when writing "currently", so we have replaced this reference by*

*Krotkov, N. A., Lamsal, L. N., Celarier, E. A., Swartz, W. H., Marchenko, S. V., Bucsela, E. J., Chan, K. L., Wenig, M., and Zara, M.: The version 3 OMI $NO_2$ standard product, Atmos. Meas. Tech., 10, 3133-3149, https://doi.org/10.5194/amt-10-3133-2017, 2017.*

Figure 4: please define or omit "golden day"

*We have replaced in the figure caption 'golden day' by 'AROMAT campaign', as the mobile DOAS and SWING measurements were always performed around these same places during the campaign.*

Technical corrections:

P1., line 14 and ff: "molec.cm-2" Is this properAMT style or should it be molec. x cm-2?

*The Copernicus guidelines only state that "Units must be written exponentially (e.g. W $m^{-2}$)". We removed the dot from our initial formulation, writing: 'molec cm-2'. This format is used by many other published studies in AMT. All the units were changed across the paper to follow this convention.*

p.2, line 4: "wingspan wider than"

*Done.*

p.5., line 10: "280m" is missing space

*Done.*

p.7, line 5: remove ")"

*Done.*

---

## Author Comment (AC2) · 9 Dec 2017

*We first thank the referee for his review and for his constructive remarks which were used to improve the manuscript. Our replies to the referee comments are in italic, the changes in the manuscript are in bold.*

1. The important issue of VCD calculations is the AMFs. During the flight, the AMFs for SWING also depend on viewing angle, geometric angle, i.e. roll and pitch etc. Are these factors considered in the calculation of AMF?

*We thank the reviewer for pointing this out as this was not properly described in the paper. In practice we have built a look up table of AMFs on a grid of the effective viewing angles (taking into account the attitude) and we have interpolated on it to estimate the AMF. This is why we had written 'for each spectrum' p9 L 8. This was made more explicit in section 4.3 of the text.*

**The viewing angle varies for each spectrum due to the scanning and the variations in UAV attitude. This is accounted for by interpolating the AMFs on a look-up table built on a 10° grid of the viewing angle.**

2. The saturated absorption effect of water vapor may bring errors to the spectral retrieve of water vapor. Do the authors consider about this?

*The SCD of water vapor does not vary linearly with respect to its column since its fine spectral structures are generally not resolved by UV-vis spectrometers, this is investigated e.g. in:*

*Wagner, T., Heland, J., Zöger, M., and Platt, U.: A fast $H_2O$ total column density product from GOME – Validation with in-situ aircraft measurements, Atmos. Chem. Phys., 3, 651-663, https://doi.org/10.5194/acp-3-651-2003, 2003*

*We have neglected this effect since in our case, the H2O SCD is known to be relatively small from the balloon measurements and the fact that we only use the zenith spectra. However, it is important to mention it for future applications where we would use more angles of the instruments*

**Note that the DOAS retrieval of H2O is complicated by the non-linearity due to unresolved absorption lines. This leads to a saturation effect which becomes important at high SCDs, as investigated in Wagner (2003). We have neglected this effect considering that we were only using zenith measurements in an area with relatively low $H_2O$ VCDs. The latter can be estimated from the balloon measurements to be around 9 x 10^22 molec cm^-2. The associated error could be around 5%. In future SWING $H_2O$ measurements, this assumption should be checked especially if using off-zenith angles and when comparing with ancillary observations.**

*This was also mentioned as an error source which could possibly bias negatively the measured relative humidity in section 5.2*

**As discussed in Sect.4.1, these estimated humidifies could be negatively biased by a few percent due to saturation of water vapor. They nevertheless appear realistic.**

3. SO2 is also another main product of exhaust plume from power plant. Since the spectral range of 200-750nm, it would be interesting to have a try to retrieve SO2 ?

*We tried to retrieved SO2 in our spectra but there was not enough signal in the UV with this spectrometer. This was improved in a follow up campaign (AROMAT-2) when we changed the spectrometer and had a good signal of SO2 which also enabled us to produce a SO2 map. This information was added to the new conclusion, and the AROMAT-2 experiments will be presented in the AROMAT overview paper in preparation.*

4. The middle panel in Figure 6 presents the potential temperature profiles. However, the data show the temperature increases with height. Is it thermal inversion? Inversion is easy to accumulate pollutants. What's the impact on the distribution of emissions of NO2?

*Considering the main plume, its source is very close to the observations so it is not well mixed in the boundary layer and thus we cannot use it to conclude on the effect of the inversion. But Fig 6 (left panel) also shows a second NO2 plume at a higher altitude and on both balloon flights, close to the boundary layer. This second plume may originate from another power plant in the south. This is discussed in detail in section 3.3 L. 24-31. We had initially written that the second plume was 'just above the boundary layer'. It could actually be at the top of the BL, trapped in the capping inversion. It is however difficult to state that with certainty with the data. We have reformulated the text to mention this.*

**Note that the elevated NO2 layer detected on both soundings appears close to the capping inversion, particularly visible in the morning balloon flight.**

5. The Mobile DOAS VCDs are significantly higher (by approximately a factor of 2) than the airborne VCDs. Why? Please give some discussions.

*This discrepancy is discussed p13 l.1 to 15, when we present Fig. 13. Actually, the Mobile DOAS VCDs are higher inside the plume but the difference shrinks and even gets opposite outside of the plume. Beside errors on AMF inputs and time differences, we mention the smoothing effect due to the radiative transfer for the airborne instrument. We have added some numbers to be more quantitative with this explanation*

**At the time of the $NO_2$ mapping flight, the solar zenith angle was close to 45°. Assuming the $NO_2$ plume to be 800 m thick, as the balloon flight indicates, leads to a horizontal smoothing of the same length for the airborne observations. Considering the sharp gradients of the NO2 field close to the power plant, this smoothing probably explains a major part of the difference between the aircraft and the mobile DOAS measurements.**

6. Is there any temperature control for the instrument (spectrometer)? For the ascent and descent, how about the spectral stability?

*There is no active temperature control, however during operation, the SWING box is heated by the PC-104. The temperature effect on spectra is partly dealt with by fitting a pseudo cross-section corresponding to change in spectral resolution, as described in section 4.1 (p. 7; l.26 -28).*

7. The spectral resolution of SWING is 1.2nm or 1.3nm? 1.2nm in table 1, but 1.3nm on Page 3, Line 22.

*The spectral resolution is 1.2 nm, the 1.3 nm on p. 3 was corrected.*